# Structural color three-dimensional printing by shrinking photonic crystals

Yejing Liu[1], Hao Wang[1], Jinfa Ho [2], Ryan C. Ng[3], Ray J.H. Ng[1,2], Valerian H. Hall-Chen[4], Eleen H.H. Koay[2], Zhaogang Dong [2], Hailong Liu[1], Cheng-Wei Qiu [5], Julia R. Greer [3] & Joel K.W. Yang [1,2]*

The coloration of some butterflies, Pachyrhynchus weevils, and many chameleons are notable examples of natural organisms employing photonic crystals to produce colorful patterns. Despite advances in nanotechnology, we still lack the ability to print arbitrary colors and shapes in all three dimensions at this microscopic length scale. Here, we introduce a heat-shrinking method to produce 3D-printed photonic crystals with a 5x reduction in lattice constants, achieving sub-100-nm features with a full range of colors. With these lattice structures as 3D color volumetric elements, we printed 3D microscopic scale objects, including the first multi-color microscopic model of the Eiffel Tower measuring only 39 μm tall with a color pixel size of 1.45 μm. The technology to print 3D structures in color at the microscopic scale promises the direct patterning and integration of spectrally selective devices, such as photonic crystal-based color filters, onto free-form optical elements and curved surfaces.

[1] Engineering Product Development, Singapore University of Technology and Design, Singapore 487372, Singapore. [2] Nanofabrication Department, Institute of Materials Research and Engineering, Singapore 138634, Singapore. [3] Division of Engineering and Applied Science, California Institute of Technology, Pasadena, CA 91125, USA. [4] Rudolf Peierls Centre for Theoretical Physics, University of Oxford, Oxford OX1 3PU, UK. [5] Department of Electrical and Computer Engineering, National University of Singapore, Singapore 117583, Singapore. *email: joel_yang@sutd.edu.sg

Realizing the full potential of three-dimensional (3D) photonic crystal structures with wide-ranging applications in integrated optical components[1–3], 3D photonic integrated circuitry, anti-counterfeiting security labels[4,5], and dye-free structural color printing requires the ability to pattern and position these crystals deterministically. However, this ability to position such structural colors at will has eluded us thus far. The fabrication of such 3D structures still remains a challenge, involving manual stacking of 2D structures with stringent alignment requirements[6]. Additive manufacturing via 3D printing removes the need for this cumbersome assembly process, thus enabling the deterministic fabrication of complex 3D photonic structures. These structures enable control of the optical path, polarization, and amplitude of light at the sub-microscopic scale, resulting in enhanced or new optical properties that are not observed in naturally occurring materials[4,7–13].

The lattice constants of photonic structures made by direct laser writing are in the micrometer length scale and operate in the infrared (IR) spectral region. To extend the operational window of these photonic devices to the UV–visible spectral range (100–700 nm), the lattice constants of the photonic crystals must be reduced accordingly, and the constituent material should have as high a refractive index as possible. Structural colors arising from photonic structures are of great interest as they do not degrade and can be printed at high resolutions compared to colors from pigments and dyes[14–18]. Unlike colloidal self-assembly approaches[16–23] and 2D full-color prints using electron-beam lithography[24–27], two-photon polymerization lithography (TPL) allows precise pattern placement in all three dimensions, enabling the production of a continuous hue of 3D structural photonic crystal colors by controlling its lattice constants.

Commercially available TPL printing techniques lack the necessary resolution to fabricate 3D photonic structures with stopbands/ bandgaps in the visible range. For instance, the Nanoscribe GmbH Photonic Professional GT can achieve ~500 nm lateral resolution and 200 nm linewidths with the proprietary IP-Dip resist. This resolution is limited by diffraction, material structural rigidity, and accumulation of below-threshold exposure in the photoresist that induces unwanted polymerization in surrounding areas. To improve the resolution of 3D TPL, stimulated emission depletion (STED) and diffusion-assisted high-resolution TPL approaches have been demonstrated, producing gyroid and woodpile photonic crystals with 290 and 400 nm lattice constants, respectively[28–31]. Nonetheless, the throughput of these approaches are low, and the processes require custom resists and complex systems comprising multiple laser sources with precise beam alignment. Currently no widely available 3D printing technique can achieve spatial resolutions better than ~300 nm with a single-wavelength laser beam and reasonable throughput. While shrinking methods with heat or aqueous solutions have shown dramatic size reductions[32–38], none have reported their potential in producing structural colors. There is thus a need to develop the ability to tune with precision the geometry of nanostructures that generate these colors, preferably with commercially available tools and materials. This increased level of precision enabled us to systematically correlate features in calculated bandstructures with measured reflectance spectra.

Here, we introduce a shrinking method (Fig. 1a) to enable the direct 3D printing of photonic crystals with stopbands in the visible range. Our approach combines the printing of 3D photonic structures using the Nanoscribe GmbH Photonic Professional GT and the IP-Dip resist, followed by a heat-induced shrinking process. To demonstrate the capability of this process, we fabricated woodpile photonic crystal structures with lattice constants as small as ~280 nm, a dimension comparable to the finest periodicities in butterfly scales. The refractive index of the cross-linked polymer increases in the process, which is desirable for widening the stopbands for structural color applications. As evidence of successfully fabricated photonic crystals, photonic stopbands appear in the visible range and the woodpile photonic crystal structures reflect vivid colors with hues dependent on their lattice constants. We produced full-color Eiffel Towers with voxel sizes as small as 1.45 μm ($x$ – $y$) by 2.83 μm ($z$). To the best of our knowledge, this is the first demonstration of a full-color 3D-printed object based on dielectric structural colors. The height of the 3D-printed Eiffel Tower model was as small as 39 μm, demonstrating the capability to design and fabricate 3D photonic crystals that are shrunk to fit specific colors. This technology would be broadly applicable to achieve compact photonic optical devices and metasurfaces, such as 3D integrated circuits on chips, and photonic polarizers on optical fibers that can precisely control the wavelengths and polarizations of the output light.

## Results

**Heat-induced photonic crystal colors.** We first investigate the ability of this process to produce structures at resolutions unachievable solely by conventional TPL. The woodpile lattice structure was chosen as its design is conveniently scripted and patterned rapidly by TPL. It consists of orthogonal grating stacks as shown in Fig. 1b, where $a_{xy}$ and $a_z$ denote the lattice constants of the woodpile structure in the lateral and axial directions, while $w$ and $h$ denote the width and height of the constituent rods. Polymeric woodpile photonic crystals with a range of $a_{xy}$ and $a_z$ were directly printed using a two-photon lithography system (Nanoscribe GmbH Photonic Professional GT) and the commercial acrylate-based photoresist IP-Dip$^{TM}$. Structurally stable woodpiles with clearly separated rods were obtained for $a_{xy}$ as small as ~475 nm (Supplementary Fig. 1a). Next, the photonic crystals were heated to 450 ± 20 °C in an Ar gas environment where the cross-linked polymers underwent time-dependent decomposition, which reduced the size of the photonic crystals by up to ~80% linear shrinkage (Fig. 1). This shrinkage allows us to produce structures with a minimum lattice constant of ~280 nm, clearly beyond the resolution limit of the TPL system. Note that not all of the structures could be successfully shrunk, e.g., structures with lattice constants less than 1.1 μm would simply coalesce into a homogeneous particle during the heating process.

In addition to shrinkage, heating also alters the effective shape of the laser writing spot. The laser writing spot at the Gaussian focal point of the TPL system is ellipsoidal, thus resulting in a vertical resolution that is ~3× worse than the lateral resolution, and line structures having an elliptical cross-section (Supplementary Fig. 2a). The heat-shrinking process effectively produces a more spherical writing spot (Supplementary Fig. 2b) allowing for a minimum $z$-axis resolution of ~380 nm, significantly below the two-photon Sparrow criterion in $z$-direction of ~500 nm. The fabrication reliability and reproducibility for smaller structures is also improved as we can pattern mechanically robust structures within a larger process window (Supplementary Figs. 2–8 for additional information on the heat-shrinking process). This concept is similar to that demonstrated in 2D with Shrinky Dinks, where structures printed using a simple desktop printer were later heat shrunk to micron length scales[39]. The printed complex gyroid and diamond lattices (Supplementary Fig. 2e, f) demonstrate the generality of the approach.

To determine the amount of shrinkage as a function of heating duration (Fig. 1m), we fabricated several structures with identical nominal parameters as shown in Fig. 1c and heated them for different durations. The structure has 12 repeat layers (48 stacks) with initial $a_{xy} = 1.57$ μm, $a_z = \sqrt{2}a_{xy}$, and comprises rods with

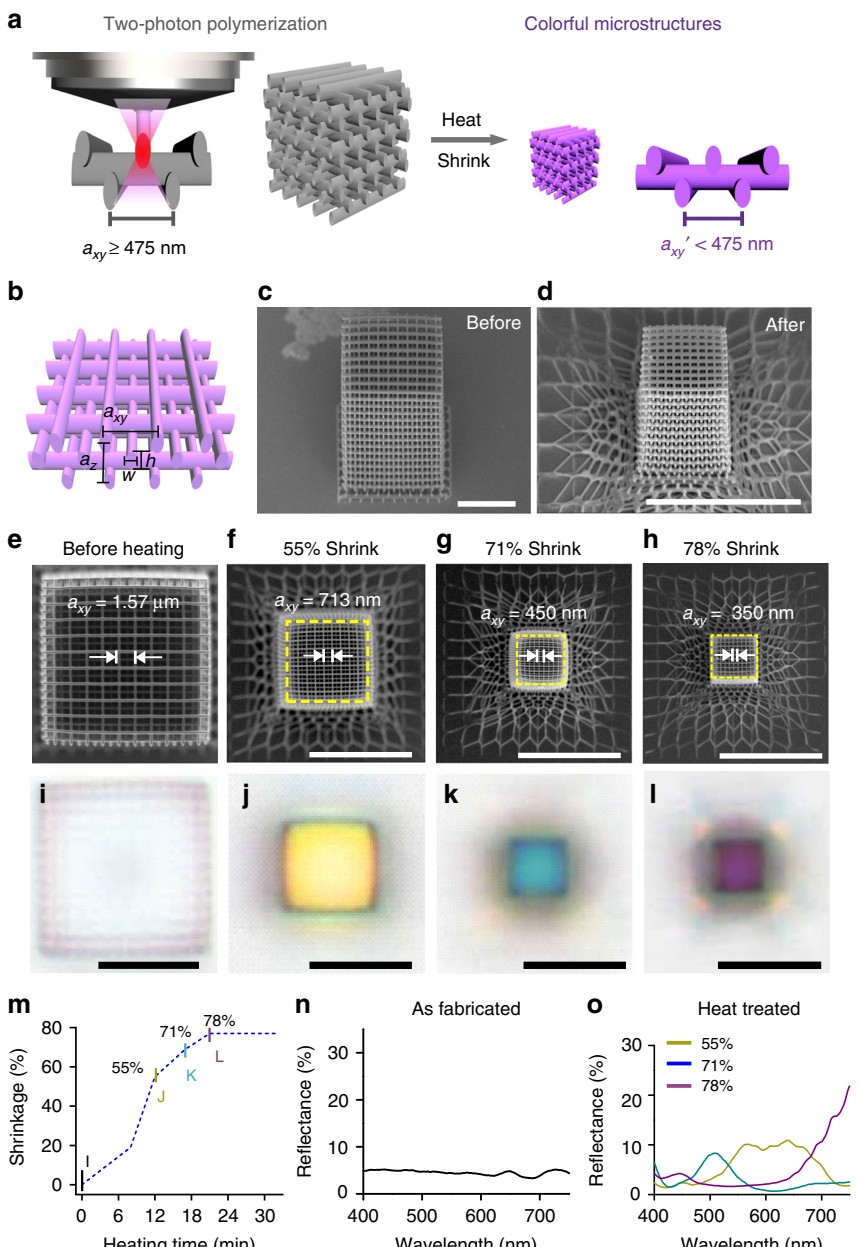

**Fig. 1** Heat shrinking induced colors of 3D-printed woodpile photonic crystals. **a** Schematic of the fabrication process. Left: woodpile photonic crystal written in commercial IP-Dip resist by two-photon polymerization at dimensions well above the resolution limit of the printer to prevent structures from collapsing. Right: after heat treatment, the dimensions of the photonic crystal are reduced below the resolution limit of the printer, and colors are generated. The colors change with different degrees of shrinkage. **b** Schematic showing one axial unit of the woodpile structure. $a_{xy}$ and $a_z$ denote the lateral and axial lattice constants, respectively. Tilted-view scanning electron micrographs (SEM) of a representative woodpile photonic crystal (**c**) before and (**d**) after heating. SEM images and corresponding brightfield reflection-mode optical micrographs of the woodpile photonic crystal before heating (**e**, **i**) and with shrinkages of 55% (**f**, **j**), 71% (**g**, **k**), and 78% (**h**, **l**). **m** Shrinkage of the woodpile photonic crystal heated at ~450 °C as a function of heating duration. Reflectance spectra of the woodpile photonic crystal **n** before heating and **o** after heating with 55%, 71%, and 78% shrinkage. Scale bars represent 10 μm

initial $w$ of 330 nm and $h$ of 1.1 μm. Figure 1e–h, i–l is SEMs and corresponding optical micrographs of the structures after 12, 17, and 21 min of heating at 450 °C, clearly showing a lateral shrinkage of 55% ($a_{xy}$ = 713 nm), 71% ($a_{xy}$ = 450 nm), and 78% ($a_{xy}$ = 350 nm), respectively. The rod width $w$ decreased to ~100 nm after 21 min of heating (Fig. 1h). After heat shrinkage, the bottom three repeat units turned into an intricate net-like mesh due to the adhesion of the bottom-most layer to the substrate (Fig. 1d). However, the top nine repeating units were observed to be uniform after shrinking as they are sufficiently far from the

substrate. The three bottom units are analogous to "rafts" in fused deposition modeling (FDM) 3D printing used to improve adhesion to the print bed, and the top nine units as a uniform photonic crystal in subsequent analyses. Alternatively, uniform structures can be achieved by printing a thick solid block underneath the structure as a buffer layer to decouple the strain mismatch between the substrate from the structure (Supplementary Fig. 6).

Next, we investigate the evolution of cross-linked IP-Dip with the heating process and propose a mechanism for the shrinkage.

From thermogravimetric analysis (Supplementary Fig. 7b), we observed that the largest reduction in mass occurred at 450 °C, thus all our samples were heated to this temperature. Raman spectroscopy measurements (Supplementary Fig. 7c) of IP-Dip before heating show the characteristic $CH=CH_2$ stretching mode at 1632 cm$^{-1}$. This peak corresponds to unreacted terminal alkene groups, indicating the presence of partially cross-linked polymers or unreacted monomers in the photonic crystal structure. After heating, the peak disappears, suggesting that these components were removed upon heating. We further observe that heating reduces the peak intensities of the $C=O$ (1722 cm$^{-1}$), C–O (935 cm$^{-1}$), and C–H (2947 cm$^{-1}$) stretching modes, while introducing peaks corresponding to activated (porous) carbon at 1593 cm$^{-1}$ (graphitic carbon), 1353 cm$^{-1}$ (disordered carbon), and 2500–3100 cm$^{-1}$ ($sp^2$-rich carbon)[40,41]. These observations indicate that the IP-Dip polymer is at the onset of carbonization, and that carbon oxide, water vapor, and small molecules of unlinked monomers are released from IP-Dip during heating. This results in structures with decreased volume and increased density and carbon content. Unlike glassy carbon that forms at higher temperatures in larger structures, the presence of the $C=O$ and C–O bands in the material after heating indicates that the material was not entirely converted into solid carbon at this temperature. Ellipsometric measurements of heat-treated IP-Dip film show an increase in the refractive index ($n$) from 1.59 to 1.82 (at 400 nm; Supplementary Fig. 7d) accompanied by an increase in the extinction coefficient ($k$) from ~0 to 0.2 (at 400 nm; Supplementary Fig. 7e), further corroborating the increase in density and carbon content in the photonic crystal structure. Compared to previous works that pyrolyzed IP-Dip at 900 °C[37], our heat treatment process achieved an almost identical amount of volume shrinkage at a much lower temperature of 450 °C. The low temperature prevents IP-Dip from turning into glassy carbon with high optical losses (with $k$ = 0.8 at 400 nm), and allows us to maintain a relatively low $k$ while increasing $n$, which is desirable for making photonic stopbands/bandgaps. Attempts to pyrolyze woodpile structures at 900 °C resulted in the complete decomposition of the woodpile (Supplementary Fig. 8). Despite the large dimensions used ($a_{xy}$ = 1.9 µm and $w$ = 403 nm), the rods could have been too thin or insufficiently cross-linked to survive the process. Heating at 900 °C therefore imposes additional limitations on the minimum rod width in order to maintain structural integrity[41].

The reduction in lattice constant and increase in $n$ of the photonic crystal resulted in colors that evolve with the degree of shrinkage. Before heating, the reflectance of the woodpile photonic crystal is weak and no colors were observed (Fig. 1i, n). Colors emerge as the structures shrink, shifting from yellow to blue and purple for linear shrinkage values of 55%, 71%, and 78% (Fig. 1j–l). The colors observed from the reflection-mode optical micrographs agree with the measured spectra (Fig. 1o), with the reflection peak center shifting from ~600 nm (55% shrinkage) to ~508 nm (71% shrinkage) and ~445 nm (78% shrinkage). At 78% shrinkage, an additional strong reflection peak was observed at ~780 nm.

The bandstructure calculations of the woodpile photonic crystals provide insight into the reflectance spectra and observed color (Supplementary Figs. 9 and 10). Without loss of generality, we consider only the bandstructures in the Γ–X and Γ–K directions, corresponding to top-down and side illumination, respectively. For the woodpile photonic crystal with $a_{xy}$ = 1.57 µm (before heating), the large number of photonic states form a continuum in the visible–IR range (Supplementary Fig. 9a) and visible light can propagate through the photonic crystal, resulting in low reflectance and a colorless appearance. After heat shrinkage to $a_{xy}$ = 350 nm, the woodpile photonic crystal exhibits

angle-dependent colors. As shown in the optical micrographs in Fig. 2a, the cubic structure appears purple from the top facet but yellow from the side. The corresponding reflectance spectra are shown in Fig. 2b with brightfield imaging configurations inset. Along the Γ–X direction of the bandstructure (Fig. 2c), a stopband is present at ~750 nm near infrared (NIR) region, corresponding to the strong reflection peak at ~780 nm observed experimentally. In addition, several states with inflection points ($i.e. \frac{d\omega}{dk} = 0$) indicative of slow light modes are present at ~430 nm. Due to impedance mismatch between the incident light and these slow light channels, coupling to these modes is poor[42–44], resulting in the reflection peak measured at ~450 nm. The structure thus appears purple under top-down illumination due to the combination of the slow light reflection peak at ~450 nm (blue) and a small spectral contribution from the tail of the strong NIR stopband reflection around 750 nm (red). The stopband along Γ–K is blueshifted relative to the stopband along Γ–X (750 nm → 705 nm) in agreement with the shift in measured reflectance peaks from the side (~75°tilt) relative to normal incidence (780 nm → 740 nm). Slow light modes were also present in the 400–450, 550–650, and 700–775 nm regions. It should be noted that the stopband at the K point is quite narrow, and the weak reflection at 740 nm under side illumination is likely due to reflection from the slow light region in close proximity to the stopband. In general, reflection from slow light mode regions tend to be weaker than reflection from stopbands as there are other bands light could couple to. Spectral features were observed at 550–650 nm in both reflectance measurements and calculated bandstructures, producing the yellow color observed under side illumination.

The reflectances of woodpile photonic crystals with varying $a_{xy}$ under normal illumination are plotted in Fig. 2d, e. For small $a_{xy}$ < 350 nm, strong peaks arise from stopbands in the NIR range and gradually blueshift into the visible spectrum as $a_{xy}$ was decreased to 300 nm (Fig. 2d). For $a_{xy}$ > 350 nm, the main peaks in the visible originate from slow light modes (Fig. 2e, Supplementary Fig. 9). A systematic redshift of the reflectance peaks is observed as $a_{xy}$ increases from 350 to 672 nm (Fig. 2f). With reflection from the slow light mode being the main determinant of the color, angle-dependent colors are observed (Supplementary Figs. 10 and 11) as slow light modes can occur at significantly different wavelengths depending on the illumination direction. The good quantitative agreement between the experimentally observed reflection peaks and the calculated bandstructure (no fitting parameters) show that stronger reflection peaks >700 nm originate from stopbands and weaker reflection peaks arise from slow light modes. While it is challenging to achieve stopbands below 700 nm using relatively low index polymers, the colors of the woodpile photonic crystals can still be tuned throughout the visible wavelength region by exploiting the reflection peak from the slow light mode.

The combination of the slow light mode in the visible and stopbands in the NIR gives rise to interesting possibilities such as tuning of the NIR reflectance peak while maintaining the same color in the visible. We fabricated woodpile photonic crystals with constant $a_{xy}$ = 450 nm but scaled the unit cell in the z-direction by introducing a factor $A$ in $a_z = A\sqrt{2}a_{xy}$, as shown in Fig. 3a. The woodpile photonic crystals in this series all appear blueish in the microscope images, but the reflectance spectra in Fig. 3b reveal an additional peak in the NIR that shifts significantly from ~800 to 1000 nm with increasing $A$. These observations agree well with the bandstructures for these photonic crystals, shown in Fig. 3c. With increasing $A$, the first Brillouin zone in the Γ–X direction becomes smaller and band folding occurs at smaller values of $k$, leading to the redshift of the stopband. Due to the

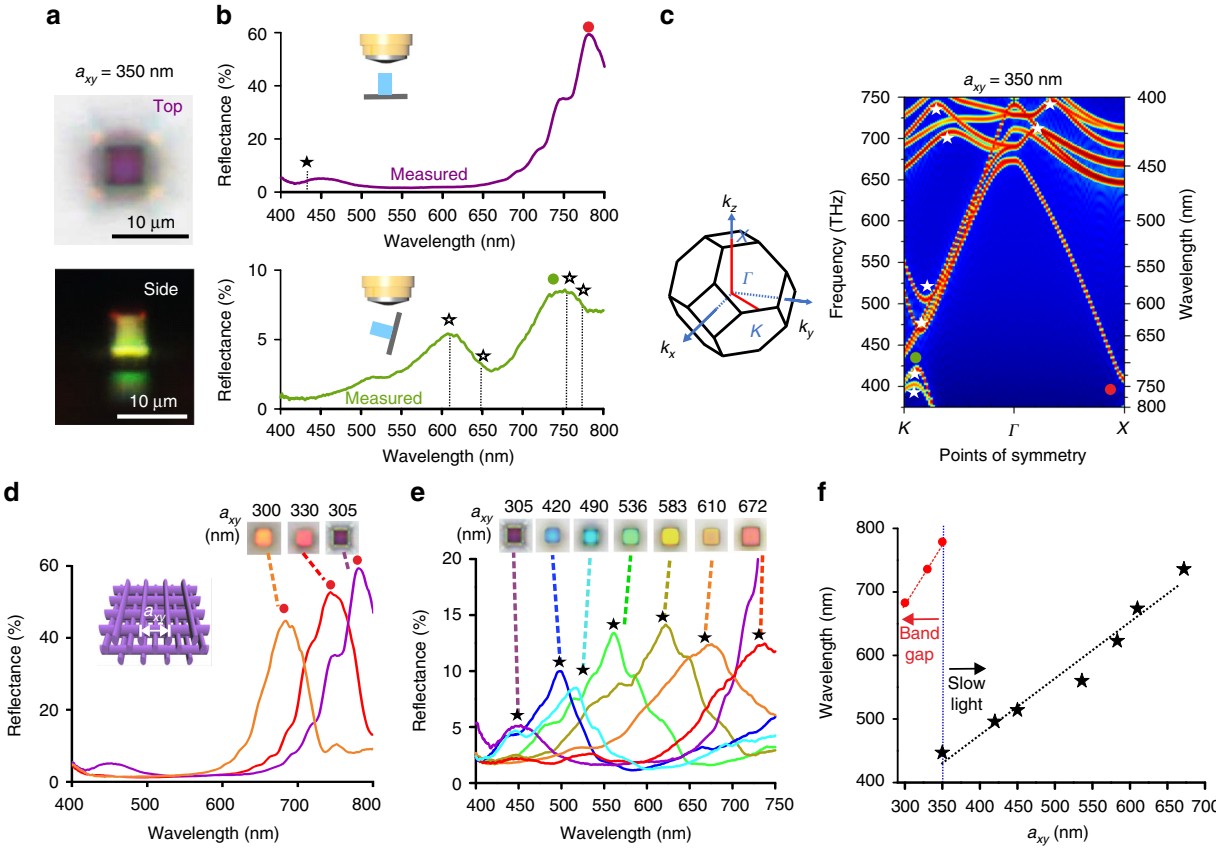

**Fig. 2** Reflectance and bandstructure of a woodpile photonic crystal with $a_{xy} = 350$ nm, $a_z = 614$ nm. **a** Top view (top) and side view (bottom) reflection-mode optical micrographs of the woodpile photonic crystal. **b** Reflectance of the woodpile photonic crystal measured with top-down illumination (top) and side illumination (bottom). **c** First Brillouin zone and photonic bandstructure of the woodpile photonic crystal in the Γ–K and Γ–X directions. Stars indicate slow light modes and dots indicate stopbands. **d**, **e** Reflectance spectra and reflection-mode micrographs of woodpiles under top-down illumination conditions for $a_{xy} = 300$–350 nm (**d**) and $a_{xy} = 350$–672 nm (**e**), respectively. **f** Plot of reflectance-peak positions as a function of the lattice constant

slow variation of the slow light mode as a function of $k$, the bandstructure for shorter wavelengths remains relatively unchanged with varying $A$, resulting in a blue hue that depends weakly on $A$. This property can be useful in encoding information in the NIR into these structures, while maintaining a constant appearance in the visible spectrum.

**3D structural color printing**. To demonstrate the printing of 3D objects consisting of structural colors at the microscale, we printed a range of woodpile structures with varying laser powers (16–27 mW) and $a_z = \sqrt{2}a_{xy}$, with $a_{xy}$ ranging from 1.1 to 2.9 μm, as shown in the composite images in Fig. 4a and Supplementary Fig. 12. As printed, these structures show little to no color (Supplementary Fig. 12a). After heating, the woodpiles are reduced in size and become colorful (Supplementary Fig. 12b), with $a_{xy}$ ranging from 330 to 980 nm, $a_z$ from 580 to 1490 nm, and $w$ from 100 to 200 nm from SEM inspection. Due to complex dependence of degree of shrinkage on laser power and pattern density, the columns in the composite image no longer have the same $a_{xy}$ but can be grouped within similar filling factors instead (Supplementary Figs. 12–14). Structures occupying the upper right side of the composite image in Fig. 4a and Supplementary Fig. 12b have the largest $a_z > 1000$ nm, thus their main reflection peaks are in the NIR region, resulting in the poorly defined colors of the woodpile structures under normal viewing. However, when viewed from the side, these structures still appear colorful due to the smaller value of $a_{xy}$, ~700 nm (Fig. 4a).

Woodpiles with different lattice constants can be freely positioned and concatenated into a single object to achieve a 3D structural color print (Supplementary Fig. 14). To demonstrate the ability to print arbitrary and complex 3D color objects at the microscale level, we fabricated microscopic models of Eiffel Towers comprised of woodpile voxels (Fig. 4). The general writing language (GWL) format layout files for the Nanoscribe were generated by filling a stereolithographic (STL) 3D model of the Eiffel Tower with woodpile structures with either constant or varying periodicities. The lattice constants were chosen to produce the desired colors after shrinking (Fig. 4c). The tower was attached to the substrate at the tip and the fabricated 3D structures are observed from the side with an optical microscope. Optical micrographs in Fig. 4 show that the Eiffel Towers have robust shapes and structures, remaining intact after thermal shrinkage, and also exhibit vivid colors. A 54 μm tall Eiffel Tower can be 3D printed either entirely in structural blue (Fig. 4d, comprising woodpile structures with $a_{xy} = 380$ nm and $a_z = 610$ nm) or structural red (Fig. 4e, comprising woodpile structures with $a_{xy} = 470$ nm and $a_z = 890$ nm), demonstrating the wide color range and versatility of our method. The woodpile structures are structurally stable and can be used as building blocks for a variety of models. To demonstrate the versatility of the method, a 20 μm tall Chinese character for luck "福" was printed in structural red (Fig. 4h, $a_{xy} = 470$ nm, $a_z = 890$ nm). Notably, in these mono-color objects, the reflected colors show little dependence on the size of the structure. We therefore estimate that the effect of Mie scattering on the colors we

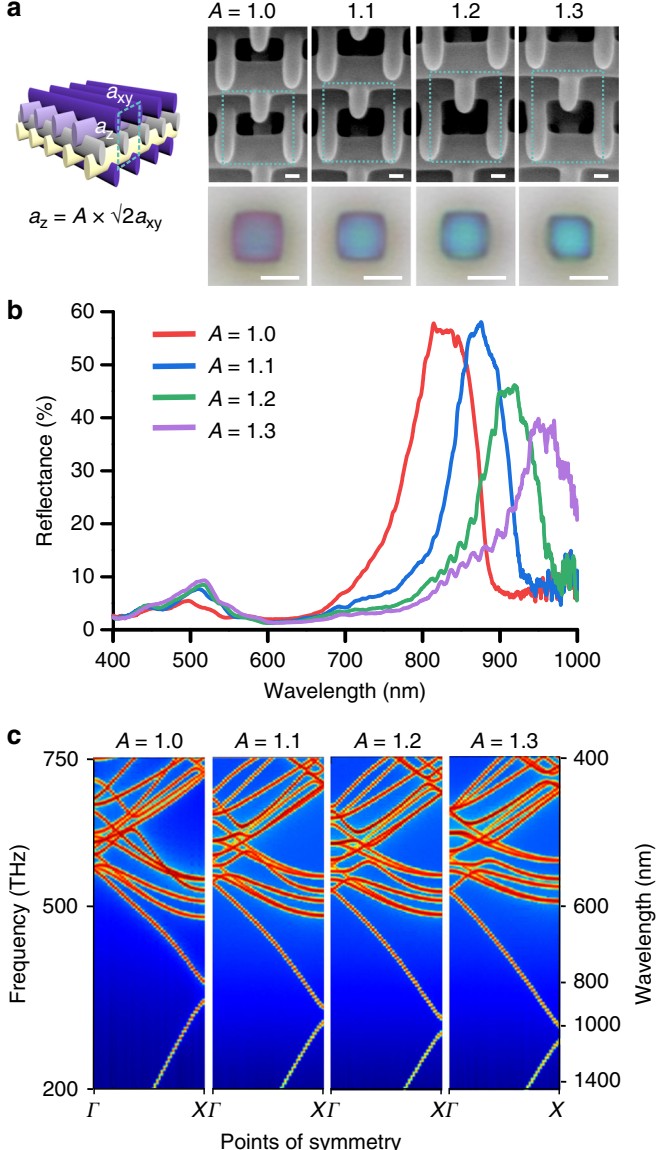

**Fig. 3** Reflectances and bandstructures of woodpile structures with fixed $a_{xy} = 450$ nm and varying $A$, the scaling factor of $a_z$. **a** 45°-tilted-view SEM and reflection-mode micrographs of the woodpile photonic crystals with $A$ varying from 1.0 to 1.3. SEM scale bars represent 100 nm and micrograph scale bars represent 5 μm. **b** Top-down reflectance spectra of the woodpile photonic crystals. **c** Bandstructures in the Γ–X direction for $A = 1.0$–1.3

observed to be minimal, as colors from Mie scattering are strongly size-dependent. Multi-colored objects can also be printed. We fabricated full-color 3D prints of the Eiffel Tower (Fig. 4f) and the ArtScience Museum in Singapore (Supplementary Fig. 15b). The fabricated Eiffel Tower 3D print had a height of 39 μm and comprises green, orange, and fuchsia color voxels (Fig. 4g). As a gauge of the color printing resolution of the woodpile structures, the smallest achievable color voxel size is 1.45 μm in the $xy$-directions and 2.63 μm in the $z$-direction (Supplementary Fig. 16).

## Discussion

The heat-induced shrinking method enables one to readily exceed the resolution limit of a 3D TPL system to print 3D objects that exhibit colors due to the underlying photonic bandstructures of the constituent lattices. The good agreement between photonic bandstructure calculations and experimental results with no fitting parameters allows us to clearly identify slow light modes and stopbands as the source of spectral peaks, giving rise to a full range of colors. While we have demonstrated that this process allows one to reproducibly create uniform or patterns of colors on a single object in cross-linked resist, this process is likely extendable to inorganic resists with higher refractive indices such as $TiO_2$ and hierarchical structures that produce angle-independent colors. Our work demonstrates the ability to produce structural color within complex 3D objects at will, and could be extended to developments in compact optical components and integrated 3D photonic circuitry that operate in the visible to NIR wavelengths.

## Methods

**Materials**. IP-Dip photoresist with a refractive index $n \approx 1.57$ (Nanoscribe Inc., Germany) was used as a negative photoresist for two-photon lithography in dip-in laser lithography (DiLL) configuration. Propylene glycol monomethyl ether acetate, isopropyl alcohol, and nonafluorobutyl methyl ether were purchased from Sigma-Aldrich. All chemicals were used without further purification. Glass slides (fused silica, 25 mm squares with a thickness of 0.7 mm) were purchased from Nanoscribe GmbH and used without further surface modification.

**Fabrication of polymeric photonic crystals on glass slides**. Polymeric nano and/or microstructures were fabricated using a direct laser writing system (Nanoscribe Inc., Germany). In a typical experiment, a droplet of IP-Dip photoresist was placed onto the bottom of a glass substrate and a microscope objective was raised into this liquid. This DiLL configuration was performed using an inverted microscope with an oil immersion lens (×63, NA 1.4) and a computer-controlled galvo stage. Predefined pattern files determine the positions of the laser spot and thus the shapes of the polymerized structures. A femtosecond pulsed laser centered at 780 nm wavelength with an average power of ~16–27 mW and a galvo-scanned writing speed of 15 mm/s were used to crosslink the resist. Unexposed photoresist was removed via immersion in propylene glycol monomethyl ether acetate for 10 min, followed by immersion in isopropyl alcohol for 5 min and nonafluorobutyl methyl ether for another 5 min. Finally, the samples were removed and left to dry under ambient condition.

**Thermal treatment of polymeric photonic crystal structures**. The polymeric photonic crystals were heated by using a temperature-controlled heating stage (Linkam Scientific Instruments Ltd). The sample was put in the closed chamber of the heating stage with Ar flow. The temperature in the chamber was heated from 26 to 450 °C at a ramp rate of 10 °C/min and was maintained at 450 °C for 17 min. After the stage heating process was stopped, the chamber was cooled down to room temperature by a water chiller.

**Bandstructure calculation**. The bandstructures were calculated using the Lumerical Finite-Difference Time-Domain software. IP-Dip was modeled as a lossless dielectric with $n = 1.57$ and $n = 1.75$ before and after heat shrinking, respectively. Electric dipoles with random orientations were placed randomly within the unit cell of the woodpile structure. Bloch boundary conditions were used, with $k_x$, $k_y$, and $k_z$ values spanning the first Brillouin zone. The electric field oscillation as a function of time was recorded, and apodized to filter out initial transient oscillations so only the oscillations belonging to the modes of the structure that propagate indefinitely remains. The fast Fourier transform of the apodized data produces the frequencies of the modes. A series of simulations with the desired $(k_x, k_y, k_z)$ values were performed to produce the bandstructures. For woodpile structures where $a_z \neq \sqrt{2}a_{xy}$, the body-centered cubic (BCC) Brillouin zone is typically used. However, for ease of comparison, we used a stretched face-centered cubic (FCC) Brillouin zone and used the same symmetry points as the FCC Brillouin zone.

**Reflectance measurement**. Optical micrographs and spectra (reflectance mode) were taken using a Nikon Eclipse LV100ND optical microscope equipped with a CRAIC 508 PV microspectrophotometer and a Nikon DS-Ri2 camera. Samples were illuminated with a halogen lamp and measured/imaged in reflection mode though a ×50/0.4 NA long working distance objective lens. The angle-varying reflectance measurements were performed on a tilt stage. The tilt angle of the stage can be tuned from 0° to 90° with 15° steps. The incident and reflected light paths were normal to the substrate when the tilt angle is 0°. The spectra (reflectance mode) are normalized to the reflectance spectrum of aluminum, which is measured under the same conditions as for the sample.

**Characterization**. Scanning electron microscopy (SEM) was performed using a JEOL-JSM-7600F SEM system with an accelerating voltage of 5 kV.

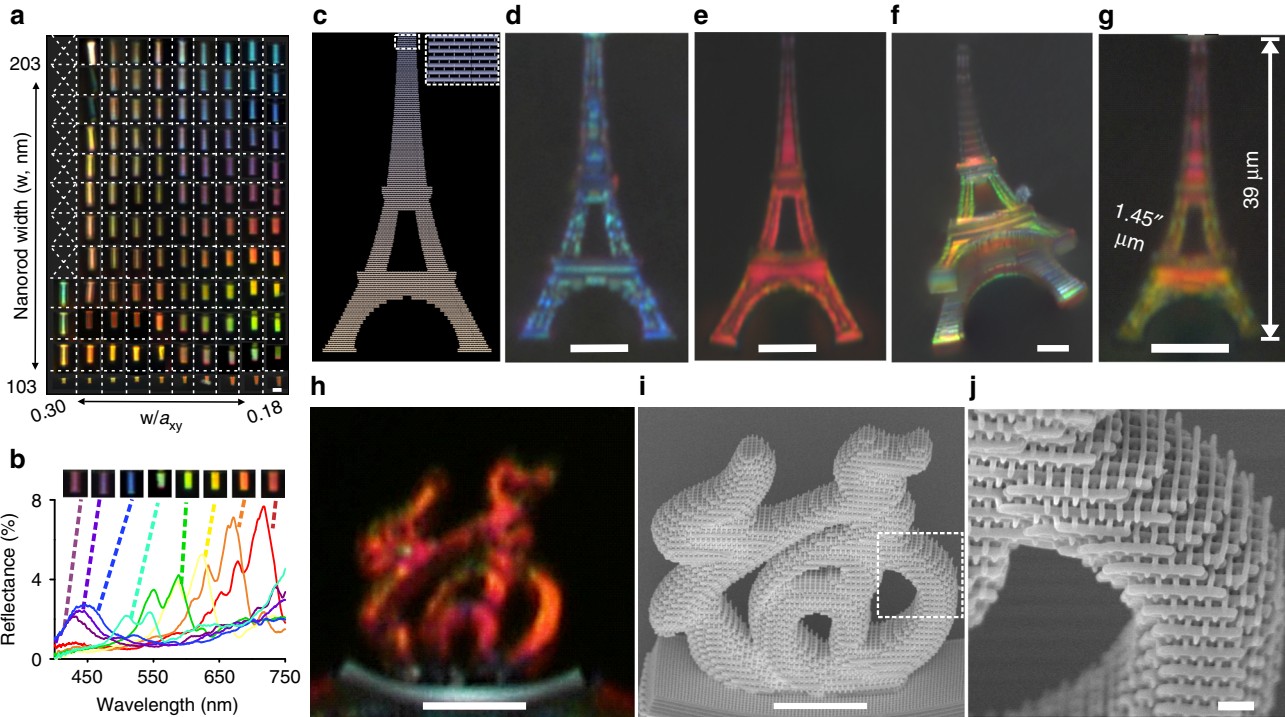

**Fig. 4** 3D color prints. **a** Composite optical micrographs of heat-treated woodpile photonic crystals with varying structural dimensions as viewed from the side. **b** Side illumination reflectance spectra of selected woodpile structures from **a**. **c** General Writing Language file used for lithographic printing of the Eiffel Tower, comprising of woodpile voxels. Micrographs of 3D-printed model of the Eiffel Tower in structural blue (**d**) and structural red (**e**). **f** Oblique view of an Eiffel Tower printed with intentional gradient of colors. **g** Further down-scaled multi-color 3D print of the Eiffel Tower. **h** Optical micrograph and **i** SEM image of a 3D Chinese character "福" in structural red. **j** Close-up SEM image of dotted square region in **i**. Scale bars in **a–i** represent 10 μm and scale bar in **j** represents 1 μm

Thermogravimetric analysis was performed using a thermogravimetric analyzer, TA Q50, with a Pt boat to hold the sample. The entire TGA analysis was performed in a closed chamber with $N_2$ flow (60 ml/min).

## Data availability

All data are available from the corresponding author upon reasonable request.

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

## Acknowledgements

J.K.W.Y. acknowledges funding support from the National Research Foundation grant awards NRF-CRP001-021 and CRP20-2017-0004, A*STAR Young Investigatorship (Grant 0926030138), and SUTD Digital Manufacturing and Design (DManD) Center grant RGDM1830303. C.-W.Q. acknowledges the financial support from the National Research Foundation, Prime Minister's Office, Singapore under its Competitive Research Program (CRP award NRFCRP15-2015-03). We express our gratitude to Robert Edward Simpson, Weiling Dong, and Tian Li for technical support with the temperature-controlled heating stage.

## Author Contributions

Y.L. designed the experiments, fabricated, and characterized the samples with assistance from H.W. and H.L. J.H. and R.C.N. performed theoretical analysis and FDTD simulations, with assistance from R.J.H.N. V.H.H.-C. fabricated samples and performed theoretical analysis. E.H.H.K. and Z.D. measured the refractive index of the IP-Dip photoresist. Structural color three-dimensional printing by shrinking photonic crystals was conceptualized by J.K.W.Y. in discussions with C.-W.Q. and J.R.G. All authors contributed to writing and revision of the manuscript.

## Competing interests

The authors declare no competing interests.
