## [Peer Review File · Nature Communications]

Reviewers' comments:

Reviewer #1 (Remarks to the Author):

The authors report on the description of a way to shrink heterostructures obtained by 3D printing of resists down to sizes where photonic effects can be observed in the visible while keeping the morphology characteristics of the initial heterostructure. The photonic effects, i.e. associated with building interferences within the periodic structure, are investigated in term of color obtained as a function of various parameters and are tentatively related to the calculated band structures.

The manuscript is well written, clear and the objectives are well stated. The subject is of broad interest, well beyond color rendering, and the topic comes timely. In particular, the possibility to use conventional resist is quite appealing. However, the present manuscript should undergo some (minor) modifications to warrant publication in Nature Communications.

The main point that should be discussed is the relation made between the band structure calculated and the optical response.

The structures shown in Figs. 1E-H are quite small with a low effective refractive index. From the resist optical constants proposed by the authors, the reader can guess that overall effective index is probably ranging between 1.2 and 1.3. Considering the total dimension of the structures observed, from 15 μ m to 4 μ m, Mie scattering should contribute significantly to the color observed. Actually, Mie scattering efficiency by spheres of these dimensions and refractive index could have the similar spectral lineshape and relative amplitude as the ones observed. Although the Mie resonances of these cubic structures are presumably different from those of spheres, could the author estimate the respective contribution to the colors observed of (1) the periodicity and (2) the overall size of the objects observed ? This could be done numerically or make use of the shape anisotropy (overall dimensions) to observe color changes under polarized light conditions, or with objectives with different numerical apertures ?

The reflection band at low wavelength is attributed to bands where dispersion is zero, referred to as slow light modes in the manuscript, associated with large DOS. There are however many other bands in the calculated band structure that could couple to light in this spectral range and many other bands with zero dispersion than the ones indicated. Moreover, large DOS does not translate necessarily to high reflectivity (see for instance Phys. Rev. B 72, 045102-2005). Could the authors discuss that point ?

The total band gap near the K point in Fig 2C, indicated by a dot in the band structure and reflectance spectrum, is not obvious. Following the description of the authors, the reflection band near 750nm could very well be associated with the so-called slow modes. The relation between the so-called slow light modes, band gap and reflectance spectra may be described more cautiously.

Other cosmetic points should be addressed

- Fig.3A, what is the scale bar on the first row ?
- The authors use sometimes the term 'maroon' (lines 198, 205...) sometimes 'purple' (lines 185, S168...) to describe the color obtained on the smallest structures. Harmonization is required. Probably 'purple' would be best appropriate since it describes the superposition of blue and red.
- The colors displayed do not always seem to match the reflection spectra measured. Fig1K shows blue hue while Fig1O shows a maximum reflection near 508nm (line 188) which should yield green color. Similar remarks hold for Fig3A. It looks like the Blue and Green canals were saturated in the color rendering.
- FigS9 : would it be possible to present only the Gamma-L direction for a better comparison with the reflectance spectra ?

Reviewer #2 (Remarks to the Author):

While this is nicely done, it is by no means the first time shrinkage is employed to increase resolution. I would like to point out the following publications, which are not mentioned in the manuscript:

1. Sun, Q., et al. (2010). "Freestanding and movable photonic microstructures fabricated by

photopolymerization with femtosecond laser pulses." *Journal of Micromechanics and Microengineering* 20(3): 035004.

2. Seniutinas, G., et al. (2018). "Beyond 100 nm resolution in 3D laser lithography - Post processing solutions." *Microelectronic Engineering* 191: 25-31.

3. Gailevičius et al., *Nanoscale Horiz.*, 2019, 4, 647-651

I do not think, therefore, that this work is novel enough for a high-impact application.

Reviewer #1 (Remarks to the Author):

The authors report on the description of a way to shrink heterostructures obtained by 3D printing of resists down to sizes where photonic effects can be observed in the visible while keeping the morphology characteristics of the initial heterostructure. The photonic effects, i.e. associated with building interferences within the periodic structure, are investigated in term of color obtained as a function of various parameters and are tentatively related to the calculated band structures.

The manuscript is well written, clear and the objectives are well stated. The subject is of broad interest, well beyond color rendering, and the topic comes timely. In particular, the possibility to use conventional resist is quite appealing. However, the present manuscript should undergo some (minor) modifications to warrant publication in Nature Communications.

We thank the reviewer for reading and recommending our manuscript for publication. And we appreciate the useful comments and suggestions.

The main point that should be discussed is the relation made between the band structure calculated and the optical response. The structures shown in Figs. 1E-H are quite small with a low effective refractive index. From the resist optical constants proposed by the authors, the reader can guess that overall effective index is probably ranging between 1.2 and 1.3. Considering the total dimension of the structures observed, from 15 μm to 4 μm , Mie scattering should contribute significantly to the color observed. Actually, Mie scattering efficiency by spheres of these dimensions and refractive index could have the similar spectral lineshape and relative amplitude as the ones observed. Although the Mie resonances of these cubic structures are presumably different from those of spheres, could the author estimate the respective contribution to the colors observed of (1) the periodicity and (2) the overall size of the objects observed ? This could be done numerically or make use of the shape anisotropy (overall dimensions) to observe color changes under polarized light conditions, or with objectives with different numerical apertures ?

Indeed, this is an interesting question. We have performed additional FDTD simulations to investigate the effect of Mie scattering. The following figure shows the FDTD simulated Mie scattering efficiencies from 3 unstructured solid cubes with refractive index of 1.25 (obtained from the estimate of the overall effective index) in free space. The side lengths are 4 μm , 6 μm and 8 μm . The wavelength dependent scattering efficiency is seen to be strongly dependent on the dimension of the cube, which is what we expect:

Based on this, we would have expected significant color differences from structures of different sizes. However, from Figures 4d, 4e, and 4F-J, we can see that the photonic crystal structures with different sizes give the same color when they have the same periodicity. Therefore, there is a stronger dependence of the color on the periodicity of the structure (i.e. bandstructure) compared to the size of the structure.

In addition, there is a strong directionality in Mie scattering. The main scattering direction is in the transmission direction. The following figure shows the scattering intensity of the 4 μm cube as functions of the wavelength and observation angle, where $\phi = 1.57$ radians is in the transmission direction:

The scattering profiles of the 6 μm and 8 μm cubes exhibit the same trend of maximum scattering in the transmission direction. As our observation of color is in the reflection, we do not expect Mie scattering to contribute significantly.

We have added a note in the second last paragraph noting the negligible contribution of Mie scattering to the observed colors:

“To demonstrate the versatility of the method, a 20 μm tall Chinese character for luck “福” was printed in structural red (Figure 4H, $a_{xy} = 470 \text{ nm}$, $a_z = 890 \text{ nm}$). **Notably, in these mono-color objects, the reflected colors show little dependence on the size of the structure. We therefore estimate that the effect of Mie scattering on the colors we observed to be minimal, as colors from Mie scattering are strongly size-dependent.** Multi-colored objects can also be printed. We fabricated full-color 3D prints of the Eiffel Tower (Figure 4F) and the ArtScience Museum in Singapore (Figure S16). The fabricated Eiffel tower 3D print had a height of 39 μm and is comprised of green, orange and fuchsia color voxels (Figures 4G). As a gauge of the color printing resolution of the woodpile structures, the smallest achievable color voxel size is 1.45 μm in the xy-directions and 2.63 μm in the z-direction (Figure S16).”

The reflection band at low wavelength is attributed to bands where dispersion is zero, referred to as slow light modes in the manuscript, associated with large DOS. There are however many other bands in the calculated band structure that could couple to light in this spectral range and many other bands with zero dispersion than the ones indicated. Moreover, large DOS does not translate necessarily to high reflectivity (see for instance Phys. Rev. B 72, 045102-2005). Could the authors discuss that point? The total band gap near the K point in Fig 2C, indicated by a dot in the band structure and reflectance spectrum, is not obvious. Following the description of the authors, the reflection band near 750nm could very well be associated with the so-called slow modes. The relation between the so-called slow light modes, band gap and reflectance spectra may be described more cautiously.

The difficulty of coupling light from the far-field to slow light modes in photonic crystals is well-established, and often attributed to either mode mismatch at the air-photonic crystal interface or impedance mismatch (see Opt. Lett. 31, 50-52(2006), J. Phys. D. 40, 2666-2670 (2007), Opt. Lett. 32, 2638 – 2640 (2007)). In Phys. Rev. B 72, 045102-2005, the theoretical approach adopted by the authors does not seem to take into account the coupling efficiencies of incident light, and therefore does not accurately predict the presence of reflection in the large DOS regions. We have added more references for this point. We agree with the reviewer that the stopband near the K point in Fig 2C is not obvious and can be associated with the slow modes. In fact, the magnitude of the reflection is quite low (< 10%) and we would expect it to be much higher if it were due to a stopband. We have modified Fig. 2B to also include the slow light mode points in the side reflectance spectrum and the main text to note these points clearly:

“In addition, several states with inflection points (i.e. $dw/dk=0$) indicative of slow light modes are present at $\sim 430 \text{ nm}$. Due to impedance mismatch between the incident light and these slow light channels, coupling to these modes is poor,³⁸⁻⁴⁰ resulting in the reflection peak measured at $\sim 450 \text{ nm}$. The structure thus appears purple under top-down illumination due to the combination of the slow light reflection peak at $\sim 450 \text{ nm}$ (blue) and a small spectral contribution from the tail of the strong NIR stopband reflection around 750 nm (red). The stopband along Γ -K is blueshifted relative to the stopband along Γ -X (750 nm \rightarrow 705 nm) in agreement with the shift in measured reflectance peaks from the side ($\sim 75^\circ$ tilt) relative to normal incidence (780 nm \rightarrow 740 nm). Slow light modes were also present in the 400-450 nm, 550-650 nm, and 700-775 nm regions. **It should be noted that the stopband at the K point is quite narrow, and the weak reflection at 740 nm under side illumination is likely due to reflection from the slow light region in close proximity to the stopband. In general, reflection from slow light mode regions tend to be weaker than reflection from stopbands as there are other bands light could couple to.**

Spectral features were observed at 550-650 nm in both reflectance measurements and calculated bandstructures, producing the yellow color observed under side illumination.

Other cosmetic points should be addressed
 - Fig.3A, what is the scale bar on the first row ?

We have added the scales in the caption of Fig 3A.

- The authors use sometimes the term 'maroon' (lines 198, 205...) sometimes 'purple' (lines 185, S168...) to describe the color obtained on the smallest structures. Harmonization is required. Probably 'purple' would be best appropriate since it describes the superposition of blue and red.

We have changed all instances of 'maroon' to 'purple'.

- The colors displayed do not always seem to match the reflection spectra measured. Fig1K shows blue hue while Fig10 shows a maximum reflection near 508nm (line 188) which should yield green color. Similar remarks hold for Fig3A. It looks like the Blue and Green canals were saturated in the color rendering.

Occasionally, we do observe that the colors rendered by the microscope does not always match the measured spectra. This happens when the color profile, white balance and saturation/brightness of the microscope, or the particular lens of the microscope changes. However, we have kept all of these settings consistent in the measurements of the many samples fabricated. As for this particular sample, it is consistent with the spectra as the peak sensitivity for green (in the tristimulus spectra of the eye) occurs at 550 nm. There is an overlap region between blue and green at ~500 nm, close to where the peak reflectance is observed.

- FigS9 : would it be possible to present only the Gamma-L direction for a better comparison with the reflectance spectra ?

The X axis of Fig S9 was labeled wrong, the L point should read X instead. We have corrected Fig S9 and removed the Gamma-X part of the band diagram:

Reviewer #2 (Remarks to the Author):

While this is nicely done, it is by no means the first time shrinkage is employed to increase resolution. I would like to point out the following publications, which are not mentioned in the manuscript:

- 1. Sun, Q., et al. (2010). "Freestanding and movable photonic microstructures fabricated by photopolymerization with femtosecond laser pulses." Journal of Micromechanics and Microengineering 20(3): 035004.*
- 2. Seniutinas, G., et al. (2018). "Beyond 100 nm resolution in 3D laser lithography - Post processing solutions." Microelectronic Engineering 191: 25-31.*
- 3. Gailevičius et al., Nanoscale Horiz., 2019, 4, 647-651*

I do not think, therefore, that this work is novel enough for a high-impact application.

We thank the reviewer for the comments, but respectfully disagree on the point regarding novelty. As the reviewer has pointed out, post processing methods, e.g. pyrolysis or dehydration with salts, has been developed and employed in recent years to shrink structures [1-4]. However, most of the work has focused on the fabrication aspects and mechanical properties of the structures [5-7]. Reports on their optical properties are still only in the infrared (not visible) spectrum [2].

The aim of our work was to add a level of precision that allows for fine periodicities (~280 nm) to be realized using a less aggressive treatment than pyrolysis, which would have damaged our structures. Our goal was to realize complex 3D prints with structural colors arising from the bandstructure of photonic crystals, which has not been reported or discussed in any of the previous work.

This new level of precision and reproducibility further enabled us to systematically track features in our bandstructure calculations that match with reflectance spectral measurements. For instance, we could correlate colors to bandstop and slow-light features in the bandstructure, with colors that appear differently when viewed from the top vs side.

To address the reviewer's concern, and highlight the novelty of our work, we have added several sentences in the manuscript as follows:

"While shrinking methods with heat or aqueous solutions have shown dramatic size reductions,³³⁻³⁹ none have reported nor investigated their potential in producing structural colors. There is thus a need to develop the ability to tune with precision the geometry of nanostructures that generate these colors, preferably with commercially available tools and materials. This increased level of precision has enabled us to systematically correlate features in calculated bandstructures with measured reflectance spectra."

- [33]. Jonušauskas, Linas, Dovilė Mackevičiūtė, Gabrielius Kontenis, and Vytautas Purlys. "Femtosecond lasers: the ultimate tool for high-precision 3D manufacturing." *Advanced Optical Technologies* 8, 3-4 (2019): 241-251.
- [34]. Sun, Quan, Saulius Juodkazis, Naoki Murazawa, Vyngantas Mizeikis, and Hiroaki Misawa. "Freestanding and movable photonic microstructures fabricated by photopolymerization with femtosecond laser pulses." *Journal of Micromechanics and Microengineering* 20, 3 (2010): 035004.
- [35]. Seniutinas, G., Anja Weber, Celestino Padeste, Ioanna Sakellari, Maria Farsari, and Christian David. "Beyond 100 nm resolution in 3D laser lithography—Post processing solutions." *Microelectronic Engineering* 191 (2018): 25-31.
- [36]. Gailevičius, Darius, Viktorija Padolskytė, Lina Mikoliūnaitė, Simas Šakirzanovas, Saulius Juodkazis, and Mangirdas Malinauskas. "Additive-manufacturing of 3D glass-ceramics down to nanoscale resolution." *Nanoscale Horizons* 4, 3 (2019): 647-651.
- [37]. Xiaoyan Li, and Huajian Gao. "Mechanical metamaterials: Smaller and stronger." *Nature Materials* 15, 4 (2016): 373.
- [38]. Bauer, Jens, Almut Schroer, Ruth Schwaiger, and Oliver Kraft. "Approaching theoretical strength in glassy carbon nanolattices." *Nature Materials* 15, 4 (2016): 438.
- [39]. Zhang, Xuan, Andrey Vyatskikh, Huajian Gao, Julia R. Greer, and Xiaoyan Li. "Lightweight, flaw-tolerant, and ultrastrong nanoarchitected carbon." *Proceedings of the National Academy of Sciences* 116, 14 (2019): 6665-6672.

REVIEWERS' COMMENTS:

Reviewer #1 (Remarks to the Author):

The authors addressed the points raised satisfactorily. The manuscript may be published.